# Self-supervised learning for analysis of temporal and morphological drug effects in cancer cell imaging data

**Andrei Dmitrenko**[1,2]                                         DMITRENKO@IMSB.BIOL.ETHZ.CH
**Mauro M. Masiero**[1,2]                                         MASIERO@IMSB.BIOL.ETHZ.CH
[1] *Life Science Zurich PhD Program on Systems Biology*

**Nicola Zamboni**[2]                                             ZAMBONI@IMSB.BIOL.ETHZ.CH
[2] *Institute of Molecular Systems Biology, ETH Zürich*

**Editors:** Under Review for MIDL 2022

## Abstract

In this work, we propose two novel methodologies to study temporal and morphological phenotypic effects caused by different experimental conditions using imaging data. As a proof of concept, we apply them to analyze drug effects in 2D cancer cell cultures. We train a convolutional autoencoder on 1M images dataset with random augmentations and multi-crops to use as feature extractor. We systematically compare it to the pretrained state-of-the-art models. We further use the feature extractor in two ways. First, we apply distance-based analysis and dynamic time warping to cluster temporal patterns of 31 drugs. We identify clusters allowing annotation of drugs as having cytotoxic, cytostatic, mixed or no effect. Second, we implement an adversarial/regularized learning setup to improve classification of 31 drugs and visualize image regions that contribute to the improvement. We increase top-3 classification accuracy by 8% on average and mine examples of morphological feature importance maps. We provide the feature extractor and the weights to foster transfer learning applications in biology. We also discuss utility of other pretrained models and applicability of our methods to other types of biomedical data.

**Keywords:** Self-supervised learning, regularized learning, time-series, distance-based analysis, classification, feature importance, explainability, interpretability, cancer research.

## 1. Introduction

Deep learning has been extensively applied to the analysis of biological images (Adam et al., 2020; Kan, 2017; Meijering, 2020; Suganyadevi et al., 2021). Learning cellular features from imaging data in an automated way, instead of designing them manually with expert knowledge, resulted in a remarkable progress across many tasks, such as classification and segmentation, object tracking and others (Moen et al., 2019).

Among many studies based on deep representation learning, Yang et al. (2020) investigated cell trajectories in the feature space along the time axis. Lu et al. (2019) exploited distance measures in the feature space to quantify similarity of cells. However, no study applied distance-based analysis of temporal drug effects using learned representations. In this study, we develop a workflow to analyze effects of anti-cancer drugs with time.

Many efforts have gone into improving interpretability of deep learning for biomedical applications (Huff et al., 2021; Singh et al., 2020). Several methods have been used to study cellular phenotypes using variational autoencoders (VAEs) and generative adversarial

networks (GANs) (Lafarge et al., 2019; Goldsborough et al., 2017). Here, we propose another way to gain insights into morphological features of cells driving drug classification. As a proof of concept, we apply it to improve classification of anti-cancer drugs and visualize image regions contributing to that improvement. Therefore, our main contributions are:

- We train a convolutional autoencoder (ConvAE) on 1M cancer cell images using random augmentations and multi-crops. We provide the source code and the model for future transfer learning applications at `https://github.com/dmitrav/pheno-ml`.

- We propose a workflow to study temporal drug effects using learned representations of images with distance-based clustering analysis.

- We propose an adversarial/regularized learning setup to improve multiclass classification of drugs and visualize morphological features driving classifier decisions.

## 2. Related work

State-of-the-art (SOTA) general purpose pretrained models (e.g., ResNet-50 trained with SwAV (Caron et al., 2021a) or DINO (Caron et al., 2021b)) are often used for transfer learning applications (Chandrasekaran et al., 2021). However, their performance may drop significantly on specific datasets such as ours (Grill et al., 2020). Models trained on biological data are available, but they are usually trained on smaller datasets. Services and tools exist to assist on biological image analysis (such as CellProfiler (Carpenter et al., 2006) or DeepImageJ (Gómez-de Mariscal et al., 2021)). However, they are not designed to handle high-throughput and often do not provide direct access to extracted feature vectors. In this work, we train ConvAE on 1M image dataset comprising 21 cell lines and 31 cancer drugs on 5 concentrations. We use random augmentations and multi-crops, prove the representations contain meaningful biological information and provide the trained model with minimal API to extract features.

A number of approaches to improve interpretability of deep learning are based on autoencoders (Biffi et al., 2020; Hou et al., 2019). Often, they are used to localize and visualize pathologies or lesions (Uzunova et al., 2019). Perhaps the closest approach to ours is the one proposed by Chen et al. (2020). The authors train a VAE on healthy subjects and then use it to detect outliers with MAP-based restoration. That is, the lesions are detected as noise in the process of image restoration. The detected regions are then visualized by calculating the difference between input and restored images. In this work, we use ConvAE as a feature extractor in a regularized learning setup to train the lens model conceptually introduced by Sajjadi et al. (2018). The resulting morphological feature importance maps are then obtained by calculating the difference between the reconstructed and the lensed images.

## 3. Data

We used advanced robotics, assay miniaturization and high-throughput imaging to acquire a dataset comprising 1M phase-contrast images covering 21 cancer cell lines exposed to 31 experimental and FDA-approved clinical cancer drugs at 5 logs of concentration, where

every condition was imaged every 2 hours for up to 6 days (**Figure 1**). Detailed description of the data is given in **Appendix C**.

## 4. Methods

### 4.1. Learning representations

We adopted a convolutional autoencoder (ConvAE) to learn image representations. We experimented with architectures to achieve good reconstruction quality and reasonable training time, as we used a single Nvidia GeForce RTX 2060 with 6GB only. We ended up with an architecture of 3 convolutional layers for encoder and decoder parts having a relatively large receptive field (maps of $32 \times 32$ pixels in the bottleneck layer). The total number of parameters stayed rather low (190k), which allowed faster training and feature extraction, as well as lower memory consumption.

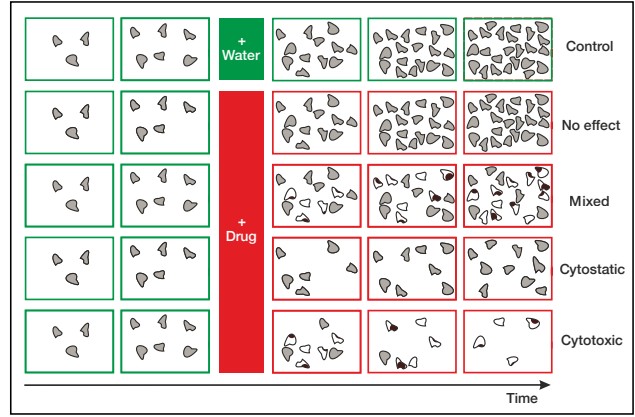

Figure 1: Schematic of the dataset and the expected drug effects over time (on the right).

### 4.2. Augmenting and cropping strategies

Since we had naturally grayscale images, we only applied random resized crops (RRCs), augmented with random Gaussian blur and horizontal flip. However, we tested a number of cropping strategies. The initial $256 \times 256$ images were randomly cropped and resized to $128 \times 128$, but the scale of RRCs varied. We tested combinations of full images and square crops of about half size and about quarter size (e.g., the following 3-crop strategy: 1 full image, 1 square crop of random size between 128-256 pixels, 1 square crop of random size between 64-128 pixels). We tested 12 cropping strategies, always having a full image and up to 4 additional RRCs of different sizes.

### 4.3. Evaluation and comparison to the pretrained models

We compared image representations obtained with ConvAE and general-purpose SOTA models pretrained on ImageNet: *i)* supervised ResNet-50, *ii)* self-supervised ResNet-50 (SwAV), *iii)* self-supervised ResNet-50 (DINO), *iv)* self-supervised ViT-B/8 (DINO). We evaluated performance of each model on 3 downstream tasks using multiple metrics.

**Similarity of biological replicates** First, we analyzed similarity of biological replicates in the latent space. For that, we picked the images of drugs at maximum concentrations and latest time points, where the strongest effect must be observable if present. We did that for each cell line and calculated distances between every pair of images of the same drug. We used the following distances to estimate similarity: Euclidean, cosine, correlation

and Bray-Curtis. Since biological replicates are expected to display the same effects, we expected the distances to be lower for those methods that capture the similarity well.

**Clustering of drug effects**    Next, we performed clustering of images within each cell line. We retrieved latent representations, reduced dimensions with UMAP (McInnes et al., 2020) and ran HDBSCAN (McInnes et al., 2017) clustering over multiple parameter sets. Since the true labels of drug effects were not available in this study, we evaluated the quality of partitions with the following metrics: percent of noise points, Silhouette score, Davies-Bouldin measure, Calinski-Harabasz index. We picked the best clustering performance over parameters sets and averaged them across cell lines.

**Classification of drugs vs controls**    Finally, we formulated a classification problem to differentiate between drugs and controls. We assigned label 1 to the images of maximum drug concentrations and label 0 to the control (no drug) images. We trained two-layer classifiers and calculated a few standard metrics: accuracy, recall, precision, specificity. The resulting setting is only weakly supervised, since some drugs did not in fact provoke any effect.

## 4.4. Analysis of temporal drug effects

To characterize temporal drug effects, we calculated distances between drug and control image representations at every time point and clustered trajectories of distances over time. More specifically, we aligned images of drugs and controls along the time axis first. Then, we retrieved their latent representations, averaged features across biological replicates and calculated distance to control for each drug at every time point. Finally, we normalized distances for each experimental condition, applied dynamic time warping (DTW) and $k$-means to cluster temporal patterns (**Figure 2**). In this setting, rapidly growing distance (fast divergence from control) is expected for immediate strong drug effect. And vice versa, low distance to control along the entire timeline is expected for no observable effect. We tested several distance metrics as in section **4.3**. For $k$-means, we incremented $k$ by 1 to find the minimum number of clusters covering the expected biological patterns.

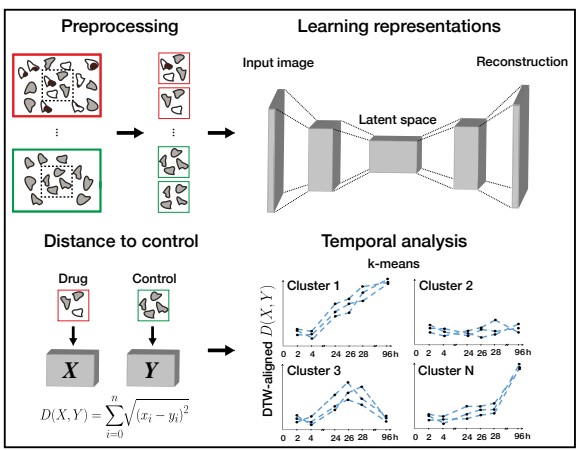

Figure 2: An overview of temporal analysis.

## 4.5. Analysis of morphological drug effects

Following the idea of shortcut removal (Minderer et al., 2020), we leveraged an adversarial learning setup to improve multiclass classification of drugs using the best pretrained model as feature extractor (**Figure 3**). The lens was trained on the images of the highest drug concentrations and the latest time points using the following loss function: $L = L_{rec} - \alpha L_{disc}$, where $L_{rec}$ is the image reconstruction loss, $L_{disc}$ is the drug discrimination loss, and $\alpha$ is an adversary coefficient. We used the same ConvAE architecture for the lens and ran grid

search for $\alpha \in [-60, 60]$, evaluating classification accuracy on the lensed images. Negative values of $\alpha$ correspond to the regularized learning.

In cases of improved classification accuracy, we visualized regions on the images perturbed by the lens. We did that by plotting the absolute difference between the lensed and the reconstructed images. The resulting regions serve as morphological feature importance maps, as they highlight regions of altered cell morphology important for drug classification.

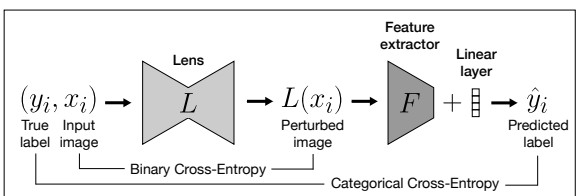

Figure 3: A schematic of the lens setup.

## 5. Results

### 5.1. Multi-crops improve performance on downstream tasks

We experimented with scales of RRCs and averaged their performance for each $n$-crop strategy, where $n \in \{2, 3, 4, 5\}$. We used multiple metrics corresponding to a particular task to average. We further normalized performance on each task, so that the top performance equals to 1.

As expected, we observed that increasing number of multi-crops improves the performance across tasks on average (**Figure 4**). However, different scales of RRCs sometimes led to sporadic drops in performance on particular tasks. The best scores across tasks were achieved by the following 5-crop strategy: 1 full size image, 1 square crop of random size between 128-256 pixels, 3 square crops of random size between 64-128 pixels. That strategy was used for training ConvAE on the entire dataset and further evaluations.

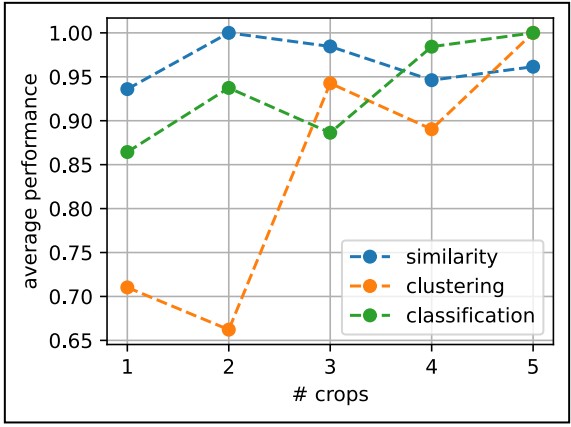

Figure 4: Normalized performance on tasks versus number of crops.

### 5.2. Comparison of pretrained state-of-the-arts

We compared several pretrained models with the ConvAE on three downstream tasks as described in section **4.3**. We report median metrics in **Table 1**.

Table 1: Comparison of pretrained models. All metrics are the higher the better.

|  | Similarity | | Clustering | | Classification | |
|---|---|---|---|---|---|---|
|  | $(\text{Euclidean})^{-1}$ | $(\text{Cosine})^{-1}$ | Silhouette | $(\text{Davies})^{-1}$ | Accuracy | F1 |
| ResNet-50 | $0.10 \pm 0.01$ | $1.21 \pm 0.03$ | $0.25 \pm 0.10$ | $0.46 \pm 0.18$ | $0.59 \pm 0.02$ | $0.59 \pm 0.05$ |
| ResNet-50 (SwAV) | $\mathbf{2.75 \pm 0.63}$ | $\mathbf{5.37 \pm 2.39}$ | $\mathbf{0.47 \pm 0.13}$ | $\mathbf{0.85 \pm 0.71}$ | $0.77 \pm 0.01$ | $0.78 \pm 0.01$ |
| ResNet-50 (DINO) | $1.07 \pm 0.02$ | $1.12 \pm 0.02$ | $0.34 \pm 0.11$ | $0.52 \pm 0.27$ | $0.72 \pm 0.00$ | $0.73 \pm 0.00$ |
| ViT-B/8 (DINO) | $2.18 \pm 0.42$ | $4.57 \pm 1.91$ | $0.44 \pm 0.12$ | $0.70 \pm 0.52$ | $0.81 \pm 0.00$ | $0.82 \pm 0.00$ |
| ConvAE (trained) | $2.26 \pm 0.68$ | $1.53 \pm 0.27$ | $0.30 \pm 0.11$ | $0.39 \pm 0.24$ | $\mathbf{0.85 \pm 0.05}$ | $\mathbf{0.85 \pm 0.05}$ |

Surprisingly, ResNet-50 pretrained on ImageNet with SwAV algorithm showed the best performance on similarity and clustering tasks. That indicates high level of consistency of the learned representations obtained with SwAV. On the other hand, the best classification accuracy (drug vs control) and F1 score were shown by our model, followed by ViT-B/8 pretrained with DINO. Therefore, features extracted by the pretrained models lacked some domain-specific information to better differentiate between drug and control images. Notably, a small model such as ours (ConvAE) can show rival performance with pretrained state-of-the-arts when trained on a large enough dataset.

### 5.3. Proof-of-concept: studying temporal drug effects

First, we analyzed representations learned by ConvAE and made sure they exhibit expected spatial and temporal separation patterns (see **Appendix B**). Then, we calculated Euclidean distance to control for each experimental condition at each time point. We further scaled, DTW-aligned and clustered the distances as described in section **4.4**.

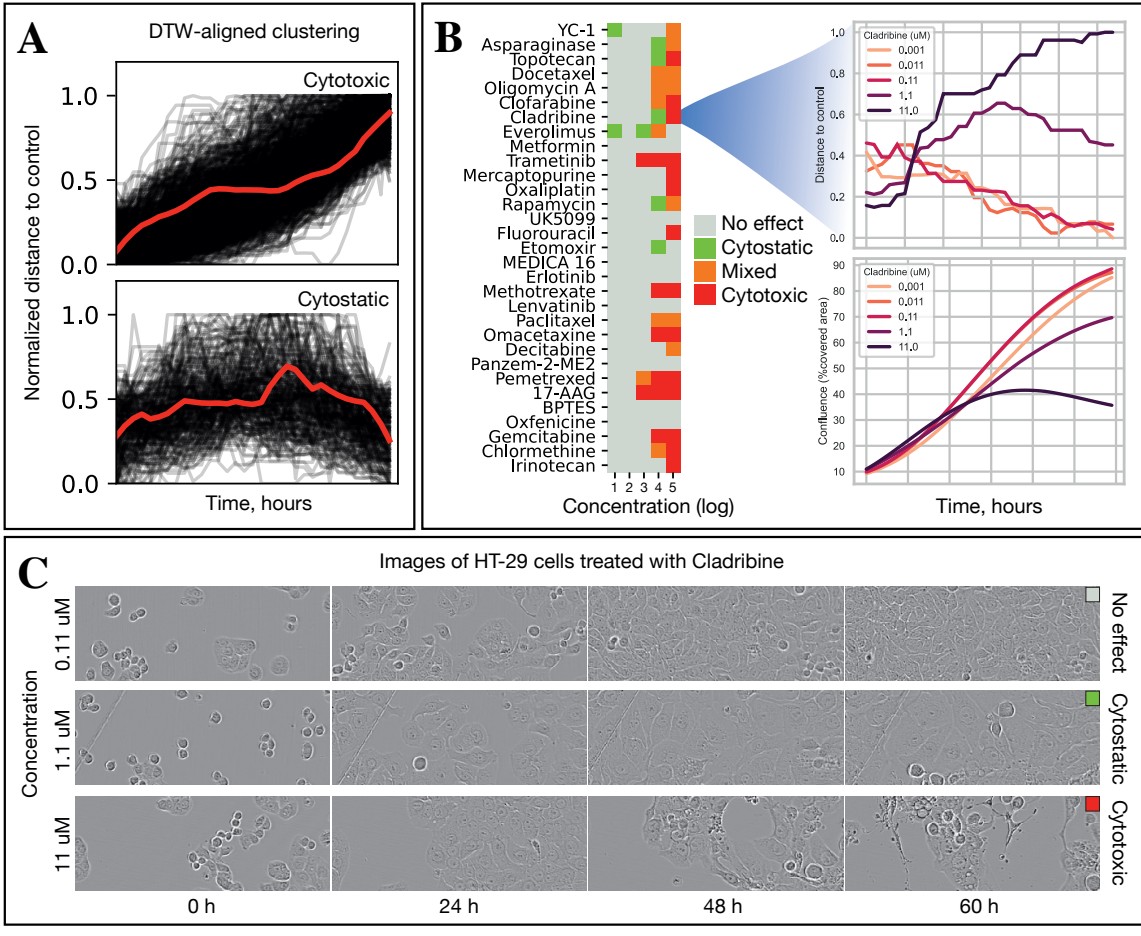

Figure 5: Analysis of temporal drug effects.

We found three clear patterns: *i)* no response, where the distance between drug-treated condition and control either stays constant or decays, so that both conditions become in-

distinguishable; *ii)* temporary response (cytostatic effect), where an initial divergence from control is observed, but ultimately reduced towards the end time points; *iii)* constant response (cytotoxic effect), where the distance grows throughout the experiment (**Figure 5A**). Red lines are mean cluster representatives. Full clustering is given in **Appendix E**.

Analyzing these patterns, we were able to annotate concentration-dependent effects for all drugs in the dataset (**Figure 5B**). Interestingly, we observed that some drugs (e.g., Cladribine) switched between cytostatic and cytotoxic modes of action, as the concentration was increased. Note that this was impossible to detect analyzing classical growth curves, since the confluence was growing for all concentrations, but the highest (**Figure 5B**, bottom-right). Conversely, distance-based analysis of learned representations allowed picking up another distinct response pattern (**Figure 5B**, top-right).

To validate such patterns, we visually inspected the corresponding images with time as shown on **Figure 5C** for HT-29 cell line and three concentrations of Cladribine. The first row shows no effect; images look identical to controls. In the last row, irregular cell morphology features (such as granules and bubbles) associated with cytotoxic effect can be seen. For 1.1 $\mu M$ concentration in the middle, we indeed observed temporary proliferation arrest accompanied with increased cell sizes. It proves that our method can distinguish between different response patterns and allows for studying temporal drug effects.

### 5.4. Proof-of-concept: exploring morphological drug effects

We used the trained ConvAE as feature extractor to train the lens, as described in section **4.5**. We observed negligible lens effects with $|\alpha| \leq 1$. Increasing $|\alpha|$ up to 60 we were able to obtain consistent improvement of top-3 classification metrics (**Figure 6**). With $\alpha = -60$, we were able to improve classification accuracy by 8%, which is significant. We looked into examples of improved classification and plotted differences between the reconstructed and the lensed images. By design, such differences highlight regions on the image that caused changes in the classification results.

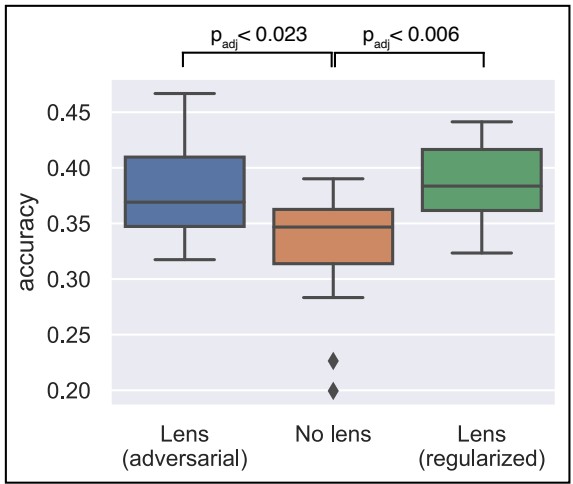

Figure 6: Multi-class classification accuracy with no ($\alpha = 0$), adversarial ($\alpha = 60$) and regularizing ($\alpha = -60$) lens.

We considered three cases of improved classification with lens as the most useful to study morphological drug effects: *i)* when the classifier initially confused a drug with a control, and the lens resolved the issue; *ii)* when the classifier initially confused a drug with another drug, and the lens resolved the issue; *iii)* when the classifier initially had low probability of correct class, and the lens dramatically increased that probability. **Figure 7** gives examples of the first two cases.

An image of Topotecan (drug) was incorrectly classified as DMSO (control) without the lens, likely due to high confluence (cell population density) on the crop (**Figure 7A**).

The output probability for DMSO was quite low though. After the lens was applied, the classifier got it right with very high confidence. Feature importance map highlights the regions of altered cell morphology that improved classification. **Figure 7B** shows an image with confluence below 50% corresponding to BPTES (drug) misclassified as Chlormethine (another drug). However, the lens identified regions of altered morphology that led to the correct classification with high confidence. We provide more examples for the increased probability case in the **Appendix A**.

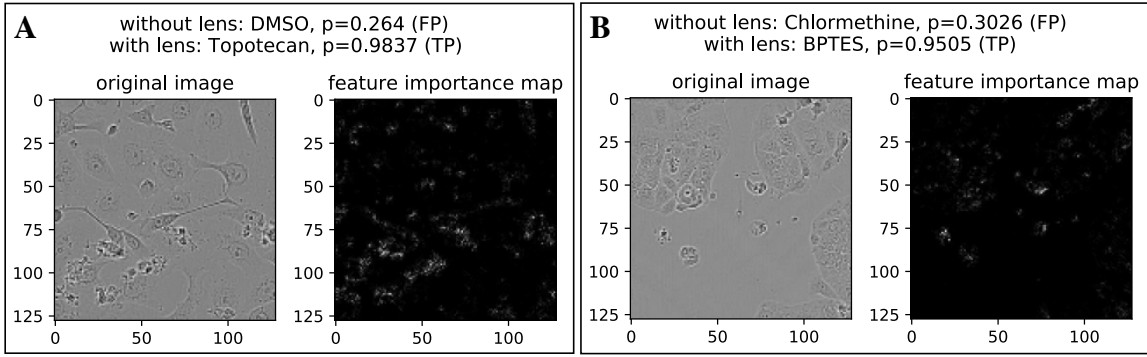

Figure 7: Classification results and morphological feature importance maps.

## 6. Discussion

The applications in **5.3** and **5.4** were done using ConvAE pretrained on a large dataset of drug-treated cancer cell lines. However, we were able to obtain temporal patterns similar to those on **Figure 5A** for individual drugs using ResNet-50 pretrained with SwAV (**Appendix D**). Thus, based on our empirical findings (**Table 1**), we speculate that ResNet-50 pretrained with SwAV could be used instead of ConvAE to study temporal and morphological drug effects with minimum information loss at no additional training cost. Although ViT-B/8 pretrained with DINO produced the closest to ConvAE classification results, using visual transformers may still be prohibitive because of their size. With Nvidia GeForce RTX 2060, we estimated the forward pass of 1M images with ViT-B/8 to take around 200 hours, with ResNet-50 – 32 hours, with ConvAE – 40 minutes.

Although we demonstrated the utility of our methods on a single biological dataset only, related works discussed earlier in this paper show comparable approaches applied to many types of biomedical imaging data. Therefore, we hope our results will contribute broadly to further development of deep learning methods for fundamental and clinical research.

## 7. Conclusion

In this work, we proposed two workflows to study phenotypic changes of experimental conditions using pretrained models. As a proof of concept, we applied them to study temporal and morphological drug effects on cancer cell lines. Besides, we trained a CNN model on a 1M images dataset comprising 21 cancer cell lines and 31 drugs at 5 concentrations. We validated the learned representations and provided the model to enable transfer learning applications. Overall, our findings suggest that pretrained models can be used for efficient and interpretable deep learning applications in biological and biomedical image analysis.

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

## Appendix A. Morphological feature importance maps

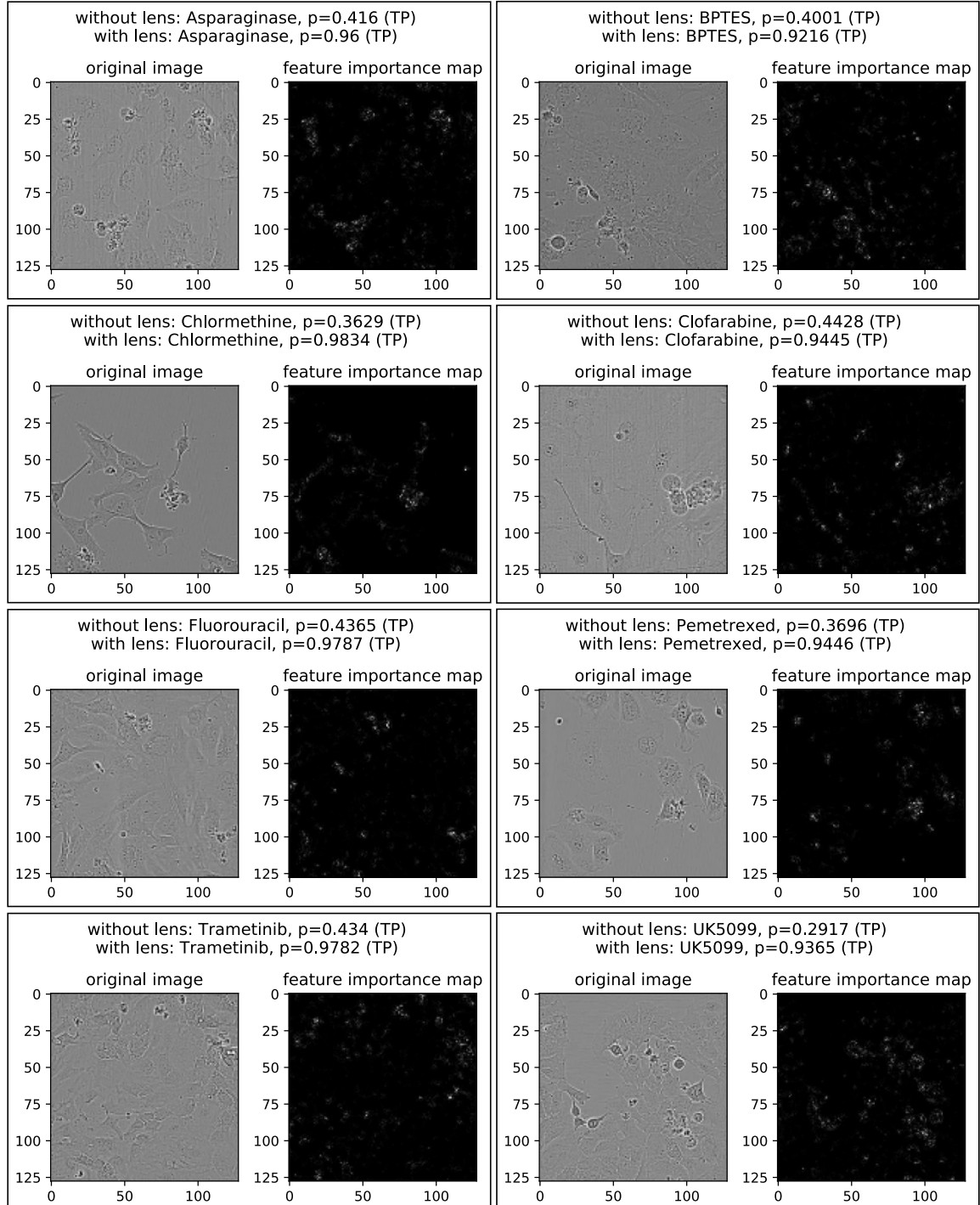

Figure 8: Morphological feature importance maps for increased classification probability.

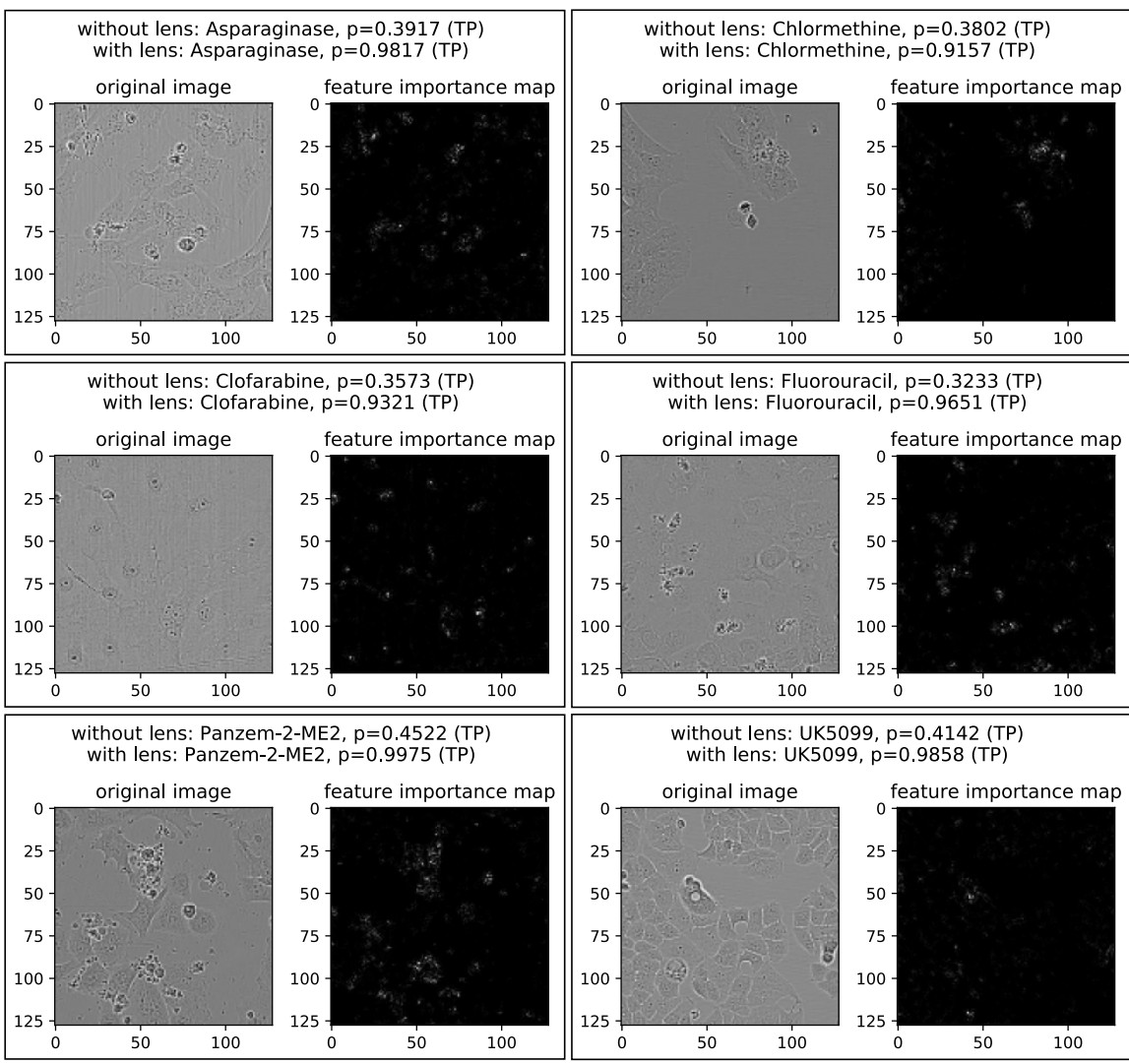

Figure 9: Morphological feature importance maps for increased classification probability.

## Appendix B. Spatial and temporal separation of UMAP embeddings

We plotted UMAP embeddings of image representations. Taking the latest time points only, we observed gradual transition from drug clusters of no effect to clusters of strong cytotoxic drug effects. Taking UMAP embeddings of all time points, we saw spatial and temporal separation of images even more clearly (**Figure 10**). Single drug tracks can be seen in colors, and some of them diverged dramatically from the initial locus (points of time $< 0$), demonstrating a variety of drug effects distinct in nature and intensity.

## Appendix C. Description of the data

To cover a wide range of phenotypic effects in experimental and FDA-approved anticancer drugs, we selected drugs that displayed at least 3 cell lines as resistant and 3 cell lines sensitive in the NCI-60 cancer cell line panel (**Table 2**), with a threshold in the $log_{10}$(GI50)

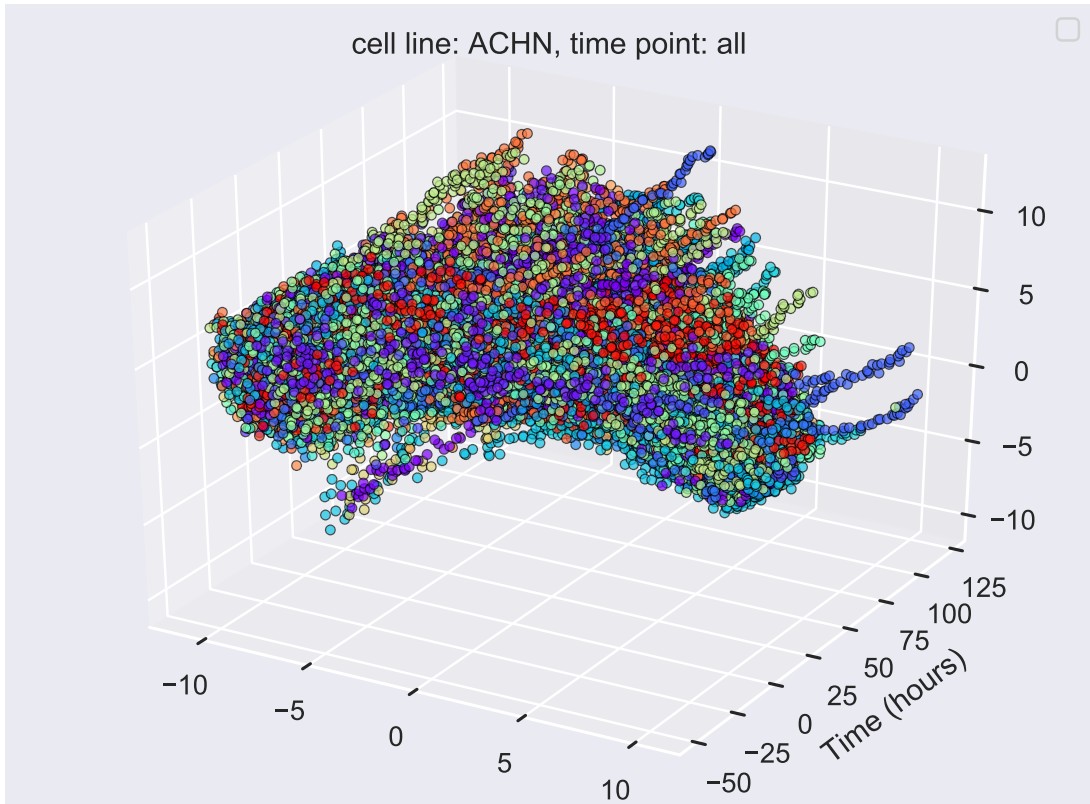

Figure 10: UMAP embeddings over time for ACHN cell line.

of 1% between the sensitive and resistant groups. The list comprised 31 experimental and FDA-approved anticancer drugs, covering several modes of action of clinical and research interest (**Table 3**).

The cancer cell lines were grown in RPMI-1640 GlutaMax medium (ThermoFischer) with supplementation of 1% of Penicylin-Streptomycin (Gibco), and 5% of dialyzed fetal bovine serum (Sigma-Aldrich) at $37°C$ in an atmosphere of 5% $CO_2$. The seeding density to achieve a confluence of 70% was determined in Nunc 96 well plates (ThermoFischer), and that seeding density was used for experiments with a factor of four correction for the reduction in area between the 96 and 384 well plates, where cells were seeded in 45 uL of medium. Cells were incubated and imaged every two hours in the Incucyte S3 (Sartorious) 10x phase contrast mode from for up to 48 hours before drug addition, in order to achieve optimal cell adherence and starting experimental conditions. To reduce evaporation effects, the plates were sealed with Breathe-Easy sealing membrane (Diversified Biotech).

To allow a broad coverage of effects on time, we collected the time information about when the drugs were treated for each cell line, and corrected the analysis based on the drug treatment. Drugs were resuspended in the appropriate solvent (DMSO or water), and the same amount of DMSO (check amount) was added across all wells, including controls. The randomized 384 drug source plates were generated with Echo Liquid Handling Sys-

Table 2: Cell lines and inoculation densities for 96 well plate format used in the study.

| Cell line | Panel | Inoculation density |
|---|---|---|
| EKVX | Non-Small Cell Lung | 11000 |
| HOP-62 | Non-Small Cell Lung | 9000 |
| COLO 205 | Colon | 15000 |
| HCT-15 | Colon | 12000 |
| HT29 | Colon | 12000 |
| SW-620 | Colon | 24000 |
| SF-539 | CNS | 10000 |
| LOX IMVI | Melanoma | 8500 |
| MALME-3M | Melanoma | 8500 |
| M14 | Melanoma | 5000 |
| SK-MEL-2 | Melanoma | 10000 |
| UACC-257 | Melanoma | 20000 |
| IGR-OV1 | Ovarian | 10000 |
| OVCAR-4 | Ovarian | 10000 |
| OVCAR-5 | Ovarian | 15000 |
| A498 | Renal | 3200 |
| ACHN | Renal | 8200 |
| MDA-MB-231/ATCC | Breast | 20000 |
| HS 578T | Breast | 13000 |
| BT-549 | Breast | 10000 |
| T-47D | Breast | 15000 |

tem (Integra-Biosciences), and then transferred in 5uL of medium to Nunc 384 well plates (ThermoFischer) with the AssistPlus liquid handler (Integra Biosciences).

Table 3: Drugs, solvents, CAS registry numbers and maximum concentrations used in the study. The other four concentrations for each drug were 10x serial dilutions of the maximum concentration.

| Drug | Fluid | CAS | Concentration |
|---|---|---|---|
| Erlotinib | DMSO | 183321-74-6 | 10 $\mu$M |
| Irinotecan | DMSO | 100286-90-6 | 10 $\mu$M |
| Clofarabine | DMSO | 123318-82-1 | 10 $\mu$M |
| Fluorouracil | DMSO | 51-21-8 | 10 $\mu$M |
| Pemetrexed | Water | 150399-23-8 | 10 $\mu$M |
| Docetaxel | DMSO | 148408-66-6 | 1 $\mu$M |
| Everolimus | DMSO | 159351-69-6 | 1 $\mu$M |
| Chlormethine | DMSO | 55-86-7 | 10 $\mu$M |
| BPTES | DMSO | 314045-39-1 | 10 $\mu$M |
| Oligomycin A | DMSO | 579-13-5 | 1 $\mu$M |
| UK-5099 | DMSO | NA | 10 $\mu$M |
| Panzem (2-ME2) | DMSO | 362-07-2 | 10 $\mu$M |
| MEDICA16 | DMSO | 87272-20-6 | 10 $\mu$M |
| Gemcitabine | Water | 122111-03-9 | 1 $\mu$M |
| 17-AAG | DMSO | 75747-14-7 | 10 $\mu$M |
| Lenvatinib | DMSO | 417716-92-8 | 10 $\mu$M |
| Topotecan | DMSO | 119413-54-6 | 1 $\mu$M |
| Cladribine | DMSO | 4291-63-8 | 10 $\mu$M |
| Mercaptopurine | DMSO | 6112-76-1 | 10 $\mu$M |
| Decitabine | DMSO | 2353-33-5 | 10 $\mu$M |
| Methothexate | DMSO | 59-05-2 | 1 $\mu$M |
| Paclitaxel | DMSO | 33069-62-4 | 1 $\mu$M |
| Rapamycin | DMSO | 53123-88-9 | 0.1 $\mu$M |
| Oxaliplatin | DMSO | 61825-94-3 | 10 $\mu$M |
| Omacetaxine | DMSO | 26833-87-4 | 1 $\mu$M |
| Metformin | Water | 1115-70-4 | 10 $\mu$M |
| YC-1 | DMSO | 170632-47-0 | 10 $\mu$M |
| Etoximir | DMSO | 828934-41-4 | 10 $\mu$M |
| Oxfenicine | DMSO | 32462-30-9 | 2.5 $\mu$M |
| Trametinib | DMSO | 871700-17-3 | 1 $\mu$M |
| Asparaginase | Water | 9015-68-3 | 0.00066 units/$\mu$L |

## Appendix D. Similar temporal patterns obtained with different models

We repeated distance-based analysis described in section **4.4** for the Cladribine case emphasized on **Figure 5B**. This time, we used representations obtained with ResNet-50 pretrained with SwAV on ImageNet. We observed many similarities between temporal patterns previously identified with ConvAE (**Figure 11A**) and the ones of ResNet-50 + SwAV (**Figure 11B**). Three lowest concentrations showed decrease of distance to control over time for both models. The highest concentration, in turns, caused constant growth of distances. For the 1.1 $\mu$M concentration, both models produced an initial increase of the distance followed by

the phase of decay. However, ResNet-50 + SwAV additionally presents an increase of the distance in the latest timepoints. This artifact is likely caused by the ImageNet biases. We, therefore, recommend to use general-purpose pretrained models to analyze more specific datasets with caution.

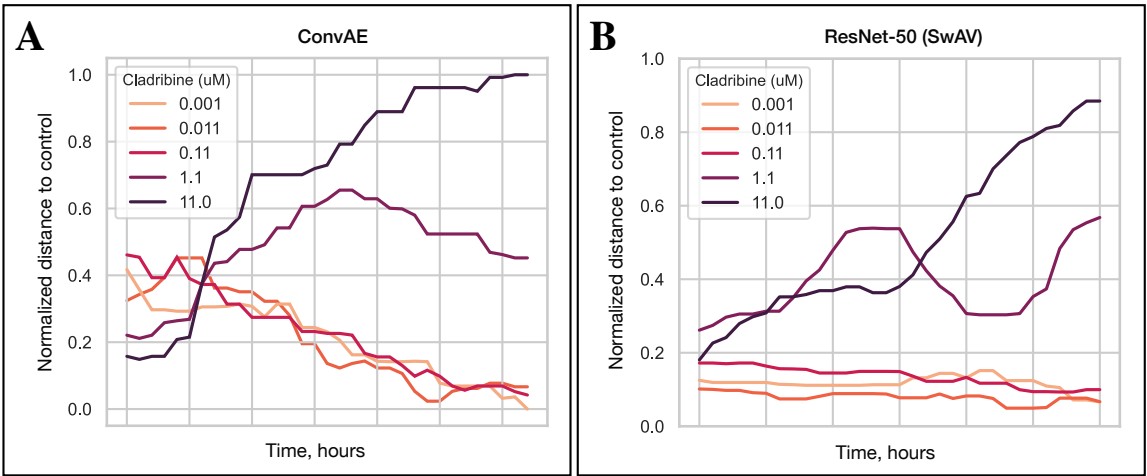

Figure 11: Temporal patterns of Cladribine obtained with different models.

## Appendix E.  Clustering of temporal patterns

In total, we found 8 clusters characterizing expected types of response, intensity and speed of divergence from controls (**Figure 12**).

We examined random cluster representatives to validate the analysis and interpret identified patterns. We found that cluster 1 is associated with a strong cytotoxic effect, as we observed a lot of cell deaths on the corresponding images. Cluster 3 represents cytostatic effect, as the images showed temporary cell growth arrest. Clusters 2 and 6 are related to mixed effects, as the images displayed both patterns.

We labeled clusters 0, 4, 5, 7 as showing no effect. Images of cluster 7 stayed indistinguishable to controls at all time points. Clusters 4 and 5 had only 30% of images showing weak cytotoxic or mixed effect.

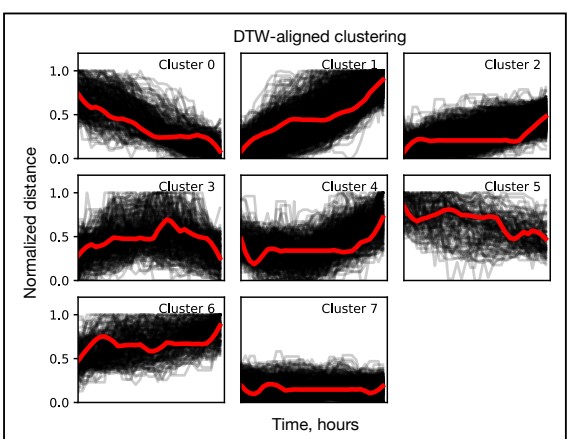

Figure 12: Clusters of temporal drug effects.

Cluster 0 is an expected artifact of the distance-based analysis: when the cell population density is low (images of early time points), the distance may be large due to varying localization of cells on the crop. With growing cell population, the distance gradually drops unless there is a drug effect.

## Appendix F. Selection of the ConvAE architecture

To be able to demonstrate novel applications in sections **5.3** and **5.4**, we needed a compact model trainable within limited resources (Nvidia GeForce RTX 2060 with 6 GB only). We tested 10 other CNN architectures and compared them by reconstruction quality. The architectures had the same number of layers as the final ConvAE, but differed in number of neurons and pooling strategies to keep the same dimensionality of the bottleneck layer. The choice of an architecture was constrained by the need to fit the data and the model into the GPU memory (especially important for the framework, described in section **4.5**). Therefore, an integration test for each architecture was performed for the lens training setup. The final architecture of ConvAE and the weights are now available on GitHub.

