# OpenReview forum: "Self-supervised learning for analysis of temporal and morphological drug effects in cancer cell imaging data"
_MIDL.io/2022/Conference — MIDL 2022_

### Official Review · Reviewer_bPGW · 2022-01-24

**Confidence:** 3
**Preliminary Rating:** 3
**Recommendation:** Poster

**Summary:**

This paper addresses the problem of cell phenotype characterisation in cancer cell images. Towards this goal, it propose a novel convolutional auto encoder model trained on 1M cancer cell images and performs comparison to four off-the-shelf pretrained models (ResNet and ViT architectures). The developed model is applied to two proof-of-concept downstream tasks: 1) temporal and 2) morphological drug effect for 31 different drugs.

**Strengths:**

- The paper is very ambitious, it not only proposes a novel model, but also compares it to four other models on surrogate performance metrics.
- The model is made publicly available, which is potentially useful for the community.
- In my view the strength of this paper is in the comparison of the different models for the two downstream tasks, and this should be emphasised more.

**Weaknesses:**

- There is a lot of focus put on the ConvAE model, which is not methodologically novel nor has particularly convincing performance on the three surrogate tasks compared to the four off-the-shelf methods.
- The other four methods are not compared for the two downstream tasks, although they might have comparative performance (this is also acknowledged in the discussion by the authors).
- It is not clear what was the experimental setup for the downstream tasks experiments. For example, setting the alpha parameter to -60 the authors claimed they achieved an improvement in accuracy of 8% but it is not clear how this apparently important parameter value was selected. Was it selected on a validation set, was there an independent test set?

**Deanonymize Review:**

no

**Detailed Comments:**

- It would be good to show more experiments (e.g. in the Appendix) about choosing the optimal ConvAE architectures.
- Please use the multiplication symbol instead of 'x' when specifying resulutions (e.g. 256×256).
- page 3: "between every 2 images of the same drug" -> "every pair of images" is more clear.
- It is unclear if the HDBSCAN clustering method was performed before or after UMAP.
- The authors state that using ViT can be computationally very expensive (200h for 1M images), however, is this really an issue in practice since 1) it linearly scales with number of GPUs and 2) it usually has to be run only once on a a limited dataset (e.g. it does not have to operate in a time-critical clinical application setting).


**Final Rating After The Rebuttal:**

4: Weak Accept

**Justification Of The Final Rating:**

In they rebuttal, the authors clarified some of the questions and concerns that I had. I believe that the revised paper manuscript has significantly improved in quality. Because of this I have updated my rating .

**Paper Type:**

validation/application paper

**Questions To Address In The Rebuttal:**

- Putting more focus on the comparison (perhaps with additional experiments for the downstream tasks of the other four methods).
- Make the experimental setup (e.g. choosing the hyper parameters) for the downstream tasks more clear.

**Special Issue:**

no

---

### Official Review · Reviewer_fzdZ · 2022-01-25

**Confidence:** 4
**Preliminary Rating:** 5
**Recommendation:** Oral

**Summary:**

The paper presents a framework that uses an autoencoder to learn latent representations to characterise cellular morphologies under different drug effect. The model is validated in a variety of different tasks, all demonstrating convincing performance including comparable performance with state-of-art pretrained models.

**Strengths:**

- Using learned representation for morphological characterisation of drug effect is an idea application area for deep representation learning. This could motivation more method development in this direction.
- The framework is tested in a variety of different tasks, demonstrating general usability.

**Weaknesses:**

- I can't find a direct correspondence between the tasks used to benchmark the pretrained models and the two downstream tasks.  Could the pretrained models perform as well in those tasks?
- The rationale for the  UMAP+HDBSCAN strategy in the clustering setting is not clear. I become even more confused when a complete different clustering strategy is used the temporal effect study.

**Deanonymize Review:**

no

**Detailed Comments:**

The last sentence in the conclusion seems strange. The paper clearly advocates the autoencoder approach rather than the pretrained models.

**Final Rating After The Rebuttal:**

5: Strong Accept

**Justification Of The Final Rating:**

The authors have addressed my concerns. I maintain my previous assessment that the paper could help enriching an application domain for fruitful methodology development. The paper has demonstrated a range of interesting tasks where deep representation learning could provide added value for quantifying cell morphologies, which could generative broad impact across cell biology.

**Paper Type:**

both

**Questions To Address In The Rebuttal:**

I'd like to the see the authors addressing the two points I listed in the weakness section. I feel the confusion could mainly due to the lack details across the experiments. I don't have any other concerns over the paper.

**Special Issue:**

yes

---

### Meta-Review · Area_Chair_P5Ek · 2022-02-15

**Recommendation:** Accept (Oral)
**Confidence:** 4

**Metareview:**

This paper has major strengths which are also outlined by the reviewers: Deep learning-based morphological characterization of drug effects, extensive evaluation, and publicly available code. The authors have also addressed the comments raised by the reviewers during their rebuttal. The topic is relevant to the MIDL conference. I believe this work would generate important discussions during the conference and should be presented at the 2022 MIDL conference.

---

### Decision · Program_Chairs · 2022-02-28

Accept